# T Cell Aging in Patients with Colorectal Cancer—What Do We Know So Far?

**DOI:** 10.3390/cancers13246227

**Published:** 2021-12-11

**Authors:** Oana-Maria Thoma, Markus F. Neurath, Maximilian J. Waldner

**Affiliations:** 1Department of Medicine 1, University Hospital Erlangen and Friedrich-Alexander University Erlangen-Nürnberg, 91052 Erlangen, Germany; Markus.Neurath@uk-erlangen.de (M.F.N.); maximilian.waldner@uk-erlangen.de (M.J.W.); 2German Center for Immunotherapy (DZI), University Hospital Erlangen and Friedrich-Alexander University Erlangen-Nürnberg, 91052 Erlangen, Germany; 3Erlangen Graduate School in Advanced Optical Technologies (SAOT), Friedrich-Alexander University Erlangen-Nürnberg, 91052 Erlangen, Germany

**Keywords:** colorectal cancer, CRC, aging, T cell aging, senescence, exhaustion

## Abstract

**Simple Summary:**

This review describes the role of T cell aging in colorectal cancer development. T cells are important mediators in cancer immunity. Aging affects T cells, leading to various dysfunctions which can impede antitumor immunity. While some hallmarks of T cell aging have been observed in colorectal cancer patients, the functional role of such cells is not clear. Therefore, understanding how aged T cells influence overall patient outcome could potentially help in the pursue to develop new therapies for the elderly.

**Abstract:**

Colorectal cancer (CRC) continues to be one of the most frequently diagnosed types of cancers in the world. CRC is considered to affect mostly elderly patients, and the number of diagnosed cases increases with age. Even though general screening improves outcomes, the overall survival and recurrence-free CRC rates in aged individuals are highly dependent on their history of comorbidities. Furthermore, aging is also known to alter the immune system, and especially the adaptive immune T cells. Many studies have emphasized the importance of T cell responses to CRC. Therefore, understanding how age-related changes affect the outcome in CRC patients is crucial. This review focuses on what is so far known about age-related T cell dysfunction in elderly patients with colorectal cancer and how aged T cells can mediate its development. Last, this study describes the advances in basic animal models that have potential to be used to elucidate the role of aged T cells in CRC.

## 1. Colorectal Cancer in Aged Individuals

Colorectal cancer (CRC) is the third most prevalent type of cancer worldwide, especially affecting the population of developed countries. According to the World Health Organization (WHO), CRC was the second most deadly type of cancer in 2020 after lung cancer [1]. The development of CRC depends on environmental factors such as diet, smoking or physical activity, as well as genetic factors, such as inherited mutations. Interestingly, age is also considered a risk factor for various types of cancers, including CRC [1,2].

Colorectal cancer incidence rapidly increases with age, where around 80% of the diagnosed cases are patients over 50 years old [3]. Furthermore, from 1988 to 2007, the number of diagnosed CRC cases over 65 years old increased by almost 40%, as data from 39 countries and 117 jurisdictions in America, Europe and Asia indicate [4]. These trends are predicted to only further increase, since the population over 65 years old continues to rise dramatically compared to the other age groups [4,5]. While the increase might remain stable in developed countries, projections on CRC incidence in developing countries describe a steep increase due to the rapidly aging population and their lifestyle [4].

CRC overall survival is also worse in aged individuals compared to younger individuals, especially in those >80 years old [6,7]. One reason for the poor outcome in aged individuals is the limited optimization of the treatment of those patients. CRC treatment is mainly based on surgical resection of malignant tumors, followed by other adjuvant therapies such as chemotherapy, radiation, or immunotherapy. While tumor resection is often used and seems to be effective independent of patient age [8,9,10], the risks associated with surgery dramatically increase with age due to the presence of comorbidities in elderly patience. In these cases (especially in patients >85 years old), surgical removal of the tumor can therefore carry greater risks than benefits [11,12]. To account for the increased comorbidities in the elderly, various scores such as Controlling Nutritional Status (CONUT) or the Charlson Comorbidity Index (CCI) can be used to determine whether a patient is suitable to receive surgery. Interestingly, CONUT status, based on general evaluation of serum albumin, total cholesterol concentration and total peripheral lymphocyte counts, shows promising results in predicting both short-term and long-term outcomes after surgery in aged patients with CRC [13]. While this might help clinicians to decide whether tumor resection in elderly CRC individuals is worth the risk, it is not a treatment.

In the realm of treatment options for elderly, chemotherapy is also a poor choice [6]. It is not clear why elderly CRC patients are less inclined to choose chemotherapy as a treatment option, since suitable aged individuals react similarly to young ones to it and the results are similar. Nevertheless, chemotherapy has countless side effects which are amplified in the elderly, and these side effects might therefore outweigh the benefits for them [6].

Immunotherapy using PD-1/PD-L1 inhibitors can be effective in treating microsatellite instable (MRI) CRC patients. Nevertheless, PD-1/PD-L1 inhibitor therapy for MRI CRC seems to be used the least in the elderly, even though the results seem to be promising across all age groups [14]. Overall, the research data on the effects on the use of such inhibitors in elderly CRC patients is very limited, especially in patients >75 years old.

Last, one very important aspect when treating elderly CRC patients is consideration of their immune system. It is well known that aging weakens the immune system, and especially the T cell panel. This phenomenon is often described as immunoaging or immunosenescence, and its impact on CRC development has barely been studied. The next sections describe the role of T cell aging in patients with CRC.

## 2. Pathophysiology of T Cell Aging

Aging is described as the loss of function at a cellular level that occurs with advancing time. It has been often correlated with malignancies such as osteoporosis [15], Alzheimer’s disease [16], heart failure [17], and colorectal cancer [3,4,5]. On a molecular level, repeated exposure to various stresses is known to lead to increased DNA damage, which promotes aging. T cells are especially predisposed to aging, since they are constantly subjected to various antigens during a lifetime. Figure 1 illustrates the changes that are usually observed during T cell aging.

### 2.1. Age-Related Changes of T Cell Receptor (TCR) Diversity

In humans, the thymic involution is an important hallmark of T cell aging. T cells mature in the thymus, where the constant interaction with major histocompatibility complex class I (MHC I) and class II (MHC II) ensures their transition into cytotoxic CD8+ and helper CD4+ T cells respectively [18]. Most importantly, their ability to extensively recognize foreign antigens comes from the diversity of T cell receptor (TCR) clones generated in the thymus. TCR consists mostly of an alpha (α) and a beta (β) chain, with around 5% of gamma (γ) and delta (δ) chains. TCR chain generation is based on the random somatic arrangements using non-contiguous variables (V), divers (D) and joining (J) gene segments [19,20,21].

The decrease in thymus function with aging deteriorates TCR clone diversity. Therefore, age-related thymic involution does not only affect the numbers of new or naïve T cells generated, but also the TCR repertoire, as illustrated in Figure 1b. Next-generation sequencing data revealed a two- to five- fold decrease in the TCR diversity in healthy aged individuals (>70 years old) compared to young ones (20–35 years old) [22]. The TCR itself also greatly changes in the elderly, where the portion that reacts to a new antigen, the so-called third complementarity determining region CDR3, is significantly shorter compared to that generated in young individuals [23].

Homeostatic clonal expansion also increases in the elderly (Figure 1b) [23,24]. A longitudinal study evaluating healthy individuals with ages between 23 to 50 years revealed that some T cell clones can persist for over 20 years, with great potential for expansion [25]. Furthermore, naïve T cell clones generated in older individuals seem to be generally less fit to develop a memory phenotype after in vitro stimulation where they underwent two or more divisions [22]. Last, with advancing age, greater clonality is observed in CD8+ T cells compared to CD4+ T cells [22].

### 2.2. Naïve/Memory Shifts and Loss of Co-Stimulatory Molecules with Age

The decrease in naïve T cells due to thymic involution is usually mirrored by an increase in the memory phenotype [24]. In short, the T cell response to an antigen prompts a shift from a naïve state usually characterized by high CD45RA and C-C chemokine receptor type 7 (CCR7) expression, towards becoming effector cells. This leads to the development of central memory (CM, CCR7+CD45RA−), effector memory (EM, CCR7-CD45RA−) and effector memory re-expressing CD45RA (TEMRA, CCR7-CD45RA+) cells [26]. Once the antigen has been removed, some memory cells will remain as protection against re-infection. Over a lifetime of active immune responses, memory T cells will predominate.

CD8+ T cells are especially prone to aging. The expression of molecules correlated with a naïve state (CCR7, CD45RA etc.) is extensively altered in elderly compared to young individuals, as shown in Figure 1a [27,28,29]. Functionally, aged CD8+ T cells cannot react to new antigens and have deficient migratory patterns [29]. Furthermore, aging also affects a very special subset of CD8+ T cells, so-called virtual memory (VM). These cells are antigen-naïve cells that show a memory phenotype to a certain degree, and their numbers are increased in the elderly. VM CD8+ T cells from young individuals are highly proliferative in response to stimuli, while aged VM CD8+ T cells proliferate poorly [28].

Similar changes occur in CD4+ T cells, where memory T cells accumulate to the detriment of naïve cells during aging [30], but to a smaller extent (Figure 1a). One hallmark specific to CD4+ T cell aging is the accumulation of regulatory T cells (Tregs) [31,32,33]. Tregs are important in regulating autoimmune responses and maintaining homeostasis. However, Treg accumulation in elderly has a highly suppressive effect in various diseases, such as neurodegenerative disorders and cancer [34,35]. Furthermore, inflammatory Th17 cells are also predominant in healthy elderly individuals (>60 years old), as compared to healthy young individuals (24–34 years old) [36].

Last, T cell aging affects the expression of co-stimulatory molecules such as CD27 and CD28 [37]. Naïve T cells express high levels of CD27/CD28, which are necessary during the response against antigens. Memory T cells, on the other hand, significantly loose these molecules, while TEMRA cells do not express them at all [38]. How CD28 expression is modulated in naïve/memory T cells is illustrated in Figure 1a. The loss of CD27/CD28 co-stimulatory molecules during aging is usually marked by an increase in the expression inhibitory molecules such as programmed cell death protein 1 gene (PDCD1 or PD-1) and cytotoxic T lymphocyte associated protein 4 (CTLA4) [38,39,40], which describes T cell exhaustion.

### 2.3. Loss of Telomeres and Telomerase Activity with Age

Telomeres are repetitive guanine-rich sequences at the end of the chromosomes meant to protect the DNA from damage. With each replication, however, the length of the telomeres becomes shorter, a phenomenon described as the “end replication problem”. The complex telomerase, which is mainly comprised of telomerase reverse transcriptase (TERT) and telomerase RNA component (TERC), is responsible for telomere synthesis [41,42]. Only a few cells, like germ or cancer cells, constantly express telomerase.

The T cell response to mitogens is characterized by multiple rounds of cell divisions. Therefore, they depend on telomerase activity to prevent extensive telomere erosion. Indeed, high telomerase activity has been observed in human T cells in response TCR and CD28 stimulation, which can remain upregulated for up to 8 days [43,44]. Nevertheless, telomere length is strongly reduced in chronically stimulated T cells such as those from patients with rheumatoid arthritis (RA) or human immunodeficiency virus (HIV) [45,46].

Telomere length and telomerase activity are also changed in aged individuals. While TERT expression and the telomerase activity of resting T cells was not different in old individuals (>40 years old) compared to young individuals (<40 years old), it is significantly reduced in the elderly upon T cell activation [44]. A longitudinal study which evaluated individuals with ages between 21–88 years old for an average of 13 years also showed that peripheral mononuclear blood cells (PMBCs) lose telomere length over time [47]. Interestingly, young individuals (<40 years old) show the highest telomere attrition amongst all age groups, suggesting that most telomere loss occurs during early adulthood. A similar phenotype is also observed in T cells that lose telomere length with age [37].

The decrease in telomerase activity is often observed in different T cell subsets. Naïve T cells usually express high telomerase activity compared to CM (medium) to EM (low) in both elderly and young individuals (see Figure 1a) [44]. Telomerase activity of CD4+ T cells in the elderly, however, shows different patterns compared to that of CD8+ T cells. CD4+ T cells from old individuals show significantly reduced telomerase activity in naïve and CM compared to those of young individuals upon activation, while old CD8+ T cells show decreased telomerase expression only in the naïve compartment. EM T cells from the elderly, on the other hand, express significantly more telomerase activity compared those from the young, but this expression is significantly less than that of naïve or CM cells [44]. Last, telomerase activity is especially increased in CD28+ T cells, which seems to maintain their telomere length well, while CD28− T cells have significantly less telomerase activity, and therefore shorter telomeres [43].

With the increase in the memory T cell compartment and loss of co-stimulatory molecules such as CD28 with age, it is not surprising that elderly individuals show deficiencies in telomere maintenance and telomere activity (Figure 1a). The importance of long telomeres and high telomerase activity is shown in high-performing centenarians, who are also better protected against age-related diseases [48].

### 2.4. Mechanisms of T Cell Proliferation, Apoptosis, and Cellular Senescence during Aging

Aging also affects T cell proliferation. As previously mentioned, the homeostatic clonal expansion of naïve T cells increases with age [23,24]. The proliferation of T cells in response to antigen encountering, however, decreases with age. This can be correlated with defective TCR signaling, as well as the loss of the CD28 co-stimulatory molecule during aging [28,49]. Progressive loss of telomeres and diminished telomerase activity also strongly impact cell proliferation [44]. Overall, T cells are less responsive to mitogens in elderly individuals, compared to young individuals.

Decreased T cell proliferation in aged individuals is also a sign of cellular senescence. Constant exposure to stimuli induces chronic stress in T cells, which leads to extensive DNA damage. Highly damaged cells will either undergo apoptosis or enter cellular senescence, characterized by an irreversible cell cycle arrest [50]. Senescent cells are therefore unable to proliferate, but remain metabolically active. Hallmarks of T cell senescence include telomere attrition, loss of CD27/CD28 co-stimulatory molecules and the accumulation of a memory phenotype [51,52]. A recent study demonstrated that CD8+ T cells from aged individuals (~60 years old) have high senescence-associated β-galactosidase (sen-β-gal) expression compared to those of young individuals (~20 years old) [53]. Increased sen-β-gal has been also correlated to changes in memory T cell accumulation and cyclin-dependent kinase inhibitor 2a (Cdkn2a or p16) upregulation, leading to cell cycle arrest.

Progressive decline of T cell function due to aging also affects apoptosis. CD4+ and CD8+ T cells from the elderly (>65 years old) are more prone to undergo apoptosis after stimulation, as compared to those from young individuals (20–29 years old) [54]. Interestingly, naïve CD45RO- T cells from old subjects die more than those from young ones in response to stimuli [55]. This phenomenon might affect the naïve/memory balance in the elderly.

### 2.5. Patterns of Cytokine Production in the Elderly

The shifts from naïve to memory cells, telomere loss and cellular senescence induction observed in T cell compartment of elderly individuals can also affect their cytokine production. For example, interleukin 2 (IL2) levels are especially changed during aging, being significantly reduced in activated T cells from elderly individuals compared to those from young ones (Figure 1c) [56].

Age-related changes are usually correlated with an altered pattern in cytokine production observed in cellular senescence, also called senescence-associated secretory phenotype (SASP). Interferon γ (INFγ) and tumor necrosis factor α (TNFα), IL6, IL4 and IL10 amongst others, are known to contribute to the pro-inflammatory phenotype of SASP [57].

Various studies have shown that the expression of INFγ and TNFα are significantly increased in in vitro stimulated T cells from old individuals (mean age >80 years old) in comparison to young ones (~28 years old) [58,59]. Interestingly, this effect is mostly predominant in the effector CD8+CD28− and memory CD8+ T cells [59,60].

Memory CD8+ T cells also show a significant increase in IL6, IL10 and IL4 production in adult (~55 years old) and old (~90 years old) individuals compared to young ones (~24 years old) [59], which might contribute to the pro-inflammatory activity often observed in the elderly (Figure 1c). Even though IL10 cytokine production is increased in aged CD8+ T cells, it is mostly characteristic to CD4+ T cells. With the accumulation of Tregs in the elderly, IL10 production can also change [32]. Interestingly, CD4+CD25+ Tregs from the elderly (>60 years old) were able to inhibit the cytokine production of CD4+CD25− T cells to a greater extent than those of young individuals (<40 years old), in an in vitro co-culture model [61].

Overall, the phenotypes observed in T cells during aging not only affect the response of elderly to infections and vaccinations [62,63,64], but also act as a predictor of frailty and mortality [65].

## 3. Importance of Adaptive Immune T Cells in CRC

Amongst all immune cell types, T cells are key players during colorectal cancer initiation, promotion, and progression [66]. Helper CD4+ T cells and cytotoxic CD8+ T cells comprise the T cell panel. CD4+ T cells are especially important as mediators of innate and adaptive immunity in cancer, including during the priming initiated by antigen-presenting cells (APCs). Their activation by APCs prompt CD4+ T cells to activate the highly cytotoxic CD8+ T cells and to migrate to the tumor site [67]. Due to their crucial role in tumor mediation, T cells are some of the most studied immune cell subtypes in CRC.

The immune landscape in patients with colorectal cancer contains both innate and adaptive immune system cells [68]. Type, number, and their activation status are modulated by the intrinsic and extrinsic signals received from all cells in the tumor microenvironment (TME) [67,69]. For example, it is well known that the numbers of T cells found at the tumor margins and in the tumor center can greatly impact the overall survival of patients with CRC. A genomic analysis and immunostaining of human CRC samples associated low disease recurrence with high CD3 immune infiltrates in the tumor margins and center, which seemed to better predict patient outcome than histology [67,70]. Nevertheless, the type of T cells at the tumor site highly impacts CRC development. Cytotoxic CD8+ T cells, type 1 helper CD4+ T cells (Th1) and follicular helper T cell (Tfh) infiltration are usually correlated with a good patient prognosis [67,70,71,72], while high numbers of type 17 helper T cells (Th17) and regulatory T cells (Tregs) in the TME or blood are not beneficial [72,73,74,75]. Most importantly, the activation status of T cells is highly relevant to the patient outcome in CRC.

In recent years, more and more research data has revealed that our understanding on the importance of T cells in colorectal cancer is incomplete. While the overall T cell numbers and subtypes infiltrated in the TME are a good indication for overall patient survival, they do not always evoke their potential in tumor cell killing. For example, not all CD8+ T cells found at the tumor site are necessarily tumor-reactive. One study has shown that CD8+ T cells collected from the blood and tumors from CRC patients undergo transitions from a naïve state to a tumor-reactive state based on their methylation patterns [76]. Furthermore, the TME is composed of both tumor reactive and so-called “bystander” CD8+ T cells, and the frequencies of such cells vary from patient to patient [76,77], which might impact their outcomes. Even though high tumor reactive CD8+ T cells were constituted by effector memory cells, they also showed a shift towards a highly exhausted phenotype, as shown by PD-1 upregulation [76]. This knowledge can be useful for immunotherapy, or even for individualized therapy, where tumor-reactive T cells could be generated by co-culturing them with patient-derived tumor organoids [78].

Interestingly, exhausted CD8+ T cells are not only especially abundant in the TME, they also have high expansion potential [79]. Similarly, TEMRA cells are highly proliferative, but their infiltration in the TME is not usually beneficial. Even though exhausted and TEMRA CD8+ T cells are shown to have low cytotoxic potential in vitro, they do not lose it completely in vivo, as shown by the high IFNγ and granzyme gene expression [79].

The expansion of CD4+ T cells is not as predominant as that of CD8+ T cells in the tumor microenvironment. Nevertheless, amongst the CD4+ T cell subtypes, TEMRA CD4+ T cells and Tregs are the most proliferative [79]. Curiously, highly suppressive Tregs are also characterized by the expression of various exhaustion markers such as PD-1 and CTLA4 [80]. This might influence the course of the disease, since high numbers of circulating CD4+CTLA4+ T cells have been associated with advanced CRC stages. The impact of T cells on CRC development is summed up in Table 1 and includes additional information on all known types of helper CD4+ T cells found in the tumor microenvironment.

Since the shift in the T cell panel from naïve to memory to highly exhausted TEMRA cells does not occur only in response to cancer but also as a hallmark of T cell dysfunction that naturally appears with age, immunoaging might highly impact the way that T cells respond to colorectal cancer.

## 4. Effect of T Cell Aging in Patients with CRC

The functional role of aged T cells in patients with colorectal cancer is not yet understood. Nevertheless, various studies describe that an aged adaptive immune system is usually correlated to a worse patient outcome. The commonly observed phenotypes of aged T cells in CRC patients refer to the naïve/memory imbalance, TCR diversity, and loss of telomeres and CD28.

Healthy aged individuals (>70 years old) without signs of extreme comorbidities usually have an adequate immune system. When comparing T cell fitness in such subjects to that of CRC patients (>70 years old), various changes are observed. Even though similar numbers of circulating CD4+ and CD8+ T cells are present in the blood, the percentages of naïve CD8+ T cells are significantly decreased in patients with CRC [82]. Decreased numbers of recently thymic emigrant CD8+ T cells are also common in these patients. The thymic output has been negatively correlated with patient age, while accumulating memory and senescent CD8+ T cells indicates a high risk of disease-free relapse [83]. This phenotype is especially predominant in stage IV CRC patients. Furthermore, immunosenescent CD8+ CD28− T cells are also frequently observed in CRC patients, which might impair the response to cancer [84].

One important hallmark of aging T cells observed in individuals with CRC was telomere loss, which was more predominant compared to that of healthy elderly individuals [82]. Interestingly, telomere erosion correlated with age in healthy elderly patients, but not in CRC patients, which suggests that immunosenescence might play an important role in cancer development. PMBC relative telomere length (RTL) can also be an indication of patient outcome in colorectal cancer. CRC patients with leukocytes with short RTL have also worse overall survival and relapse-free survival than those with long RTL [85]. Short RTL is especially seen in patients with stage III and IV CRC, but even in these patients, longer RTL is a better prognostic sign.

CD4+ T cells seem to be less affected by the same aging phenotypes as CD8+ T cells. CD4+ T cell aging is more often described as the accumulation of Tregs or immunosenescent Th17 cells. For example, elderly CRC patients (median age of 75 years old) have increased numbers of Th17 cells in the blood and at tumor site compared to age-matched healthy individuals [36]. Th17 cells are also more predominant in advanced CRC stages.

T cell aging is not only a marker for overall patient outcome in CRC, but also a risk factor in colorectal cancer development.

## 5. Potential of Using Animal Models to Elucidate the Role of Aged T Cells in CRC

Mouse models are valuable research tools that can be used for understanding the role of T cell aging. Here, we describe current wildtype and knockout models that have high potential to be used for studying the function of senescent T cells in CRC.

### 5.1. Old Wildtype Mice

Wildtype C57Bl/6J and BALB/c mice are constantly employed as control mice by researchers, in their quest to answer scientific questions. The setup of an experiment will depend on the age of these mice, since young mice (8–12 weeks old) can behave differently than older (>20 months old) mice.

The age of wildtype animals can affect everything from organ pathology to immune cell composition. For example, single cell RNA sequencing (scRNAseq) data of the cell composition in the spleens of young (7 months) and old (24 months) C57Bl/6J mice revealed decreased proportions of T cells correlated to age, which impacted CD8+ T cells more than CD4+ T cells [86]. Furthermore, CD8+ T cells from old mice proliferated less or with more delay compared to the ones from young mice after in vitro stimulation with Concanavalin A (ConA) [87]. These CD8+ T cells also express markers specific to a memory phenotype such as CD44. Old wildtype mice also accumulate more exhausted Tregs in the spleen compared to the young ones [31].

Some studies have been described a role of T cell aging in CRC using wildtype mice. Paradoxically, mature (12–15 months old) mice seem to develop smaller tumors than young mice (2–3 months old) when exposed to the subcutaneous murine carcinoma (MC38) model [88]. These mice had increased numbers of CD8+ T cells with an effector memory phenotype, but also with high integrin alpha 4 expression. This suggests a role of integrins in T cells during aging. Nevertheless, older mice with CRC react less to immunotherapy compared to young ones [89].

### 5.2. Mice with Telomere Length Deficiency

Not all aging phenotypes can be evaluated using old wildtype mice. For example, telomere length varies amongst mammal species, with humans having ~10–15 kbp and mice ~40–50 kbp at birth [90]. While mice lose telomeres 100 times faster than humans, only a 6.6% increase in short telomeres in white blood cells is observed per year [90]. Therefore, 25 month-old mice would not be representative of the telomere shortening observed the in elderly.

Several knockout mouse models can be used to study telomere loss in T cells and its impact on colorectal cancer development. For example, mice with deficiency in telomerase RNA component (TERC) and telomerase reverse transcriptase (TERT) are known to age prematurely. Telomere loss becomes evident when the mice are crossed over several generations [91,92,93]. Not much is known about how T cells behave when they critically lose telomeres in TERC^−/−^ mice. One study described that TERC^−/−^ T cells show a reduced proliferative capacity [94]. Nevertheless, late generations of TERC^−/−^ mice can develop spontaneous lymphomas, reinforcing the importance of telomeres in immune cells [95].

Recently, Blasco’s group has developed a mouse model with hyper-long telomeres [96,97]. Furthermore, telomerase gene therapy can restore the aging phenotypes in TERT-deficient mice [98]. Even though knockout animals with telomere deficiencies are excellent tools to evaluate the impact of T cell aging in CRC, there have been no studies available until recently.

### 5.3. Klotho^−/−^ Mice

Klotho protein was correlated to aging a long time ago. Mice with klotho gene deficiency show a phenotype similar to aging in humans, with a shorter lifespan than wildtype mice and a propensity to developing osteoporosis [99,100]. Interestingly, overexpression of Klotho can potentially help with treating various cancers such as colorectal or cervical carcinoma [101,102].

Klotho gene expression is also important in T cells and is correlated to age-related diseases like rheumatoid arthritis (RA) [103]. Klotho^−/−^ mice show extreme thymic involution by age of 9 weeks old, suggesting immunoaging phenotypes as well [99]. Therefore, these mice have the potential to be used to further study T cell aging in CRC models.

### 5.4. Mice with Deficiencies in DNA Repair Genes

Ku protein complex, formed by Ku70 and Ku80, is necessary during non-homologous end joining (NHEJ)-mediated DNA repair, in order to prevent extensive DNA damage. Mice deficient in Ku70, Ku80 or both exhibit early aging phenotypes and have a lifespan of ~37 weeks [104]. Similar to TERC^−/−^ mice, Ku70- and Ku80-deficient mice develop lymphomas much earlier than the wildtype mice.

In T cells, Ku70 and Ku80 function seems to vary. While Ku80^−/−^ T cells are arrested at an early developmental stage in the thymus and are not present in the spleen, Ku70 deletion does not affect TCR formation, and therefore T cells can still be found in the spleen [105]. Interestingly, Ku70-deficiency also affects inflammation in the gut. The T cells accumulate in the lamina propria of Ku70^−/−^ mice are unable to activate apoptosis (Ku70^−/−^ × p53^R172P/R172P^ mice), leading to spontaneous colitis [106]. It would be interesting to investigate if this is also leads to increased cancerogenesis.

Excision repair cross complementing gene (ERCC1) is necessary for both DNA recombination and repair. Mice with deleted ERCC protein usually die very early after birth, due to organ underdevelopment and increased premature cellular senescence [107,108]. ERCC-deficiency in hematopoietic cells using Vav1Cre^+/−^ × ERCC^−/fl^ mice allows the evaluation of premature aging in T cells without affecting the development of other organs [109]. These mice show aging-related phenotypes as early as 6 months after birth, having a predominant memory compartment and increased stress- and cellular senescence-related genes. Most importantly, the transfer of aged Vav1Cre^+/−^ × ERCC^−/fl^ splenocytes in young wildtype mice is enough to induce senescence in various organs, such as the heart, liver and brain, amongst others [109]. This suggests that an aged immune system is enough to induce systematic organ dysfunction.

### 5.5. IL10-Deficient Mice

Interleukin 10 (IL10) is produced by Tregs amongst other types of immune cells [110]. Mice with IL10-deficiency are relatively frail, have a shortened lifespan and develop spontaneous enterocolitis as early as 3–4 weeks after birth [111]. Inflammation occurrence in IL10-knockout mice has been correlated with high CD4+ and CD8+ T cells, as well as IFNγ-producing Th1 cell infiltration in the colon, suggesting a role for IL10-mediated Th1 responses in the gut [112].

As previously discussed, Treg accumulation is predominant with age. High Treg infiltration is also correlated to worse overall patient survival. Therefore, using IL10^−/−^ mice in CRC models would help in elucidating the functional role of IL10-producing Tregs in aged CRC patients.

Table 2 summarizes the animal models that have potential for basic research in studying T cell aging in CRC.

## 6. Conclusions

Aged individuals are generally frail, and are prone to develop various chronic conditions and even cancer (Figure 2). One important factor contributing to disease onset in the elderly is an altered immune system. T cells are especially impacted by age, with deficiencies ranging from decreased T cell output due to thymic involution to low effector function after antigen stimulation. Since T cells are crucial in mediating colorectal cancer development, it is not surprising that CRC mostly affects the elderly (Figure 2). Aged individuals with CRC have a low thymic output and increased numbers of immunosenescent CD8+CD28− T cells. Very often, these cells also have shortened telomeres, which might further impact T cell effector function by inducing cellular senescence and a senescence-induced secretory phenotype. Treg accumulation during aging can also be unfavorable for elderly CRC patients, since it seems to inhibit an efficient response to the disease (Figure 2).

In conclusion, aging significantly impacts individuals with CRC, with changes observed not only due to the accumulated comorbidities, but also because of the impaired immune system. Understanding the functional role of senescent T cells would significantly improve the treatment of elderly CRC patients. One way to evaluate T cell aging behaviors in CRC patients would be using single-cell RNA sequencing. Current studies provide valuable insights not only on the specific subtypes of exhausted/senescent T cells found in CRC [81,114,115], but also on immune cell distribution in colon versus rectal tumors [116]. Furthermore, using mouse models with progressive aging phenotypes could be an excellent tool to study how aged T cells impact CRC development (Figure 2).

## Figures and Tables

**Figure 1 cancers-13-06227-f001:**
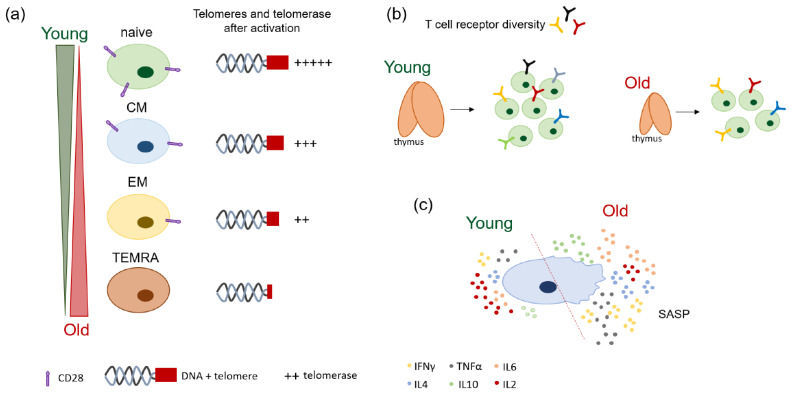
Schematic representation of age-related changes in T cells. (**a**) Accumulation of memory T cells lacking CD28 co-stimulatory molecules and with defects in telomeres and telomerase activity is predominant in the elderly. (**b**) Thymus involution is a hallmark of aging, and it reduces TCR diversity and T cell clonality. (**c**) Aging leads to senescence-associated secretory phenotype (SASP) induction in T cells.

**Figure 2 cancers-13-06227-f002:**
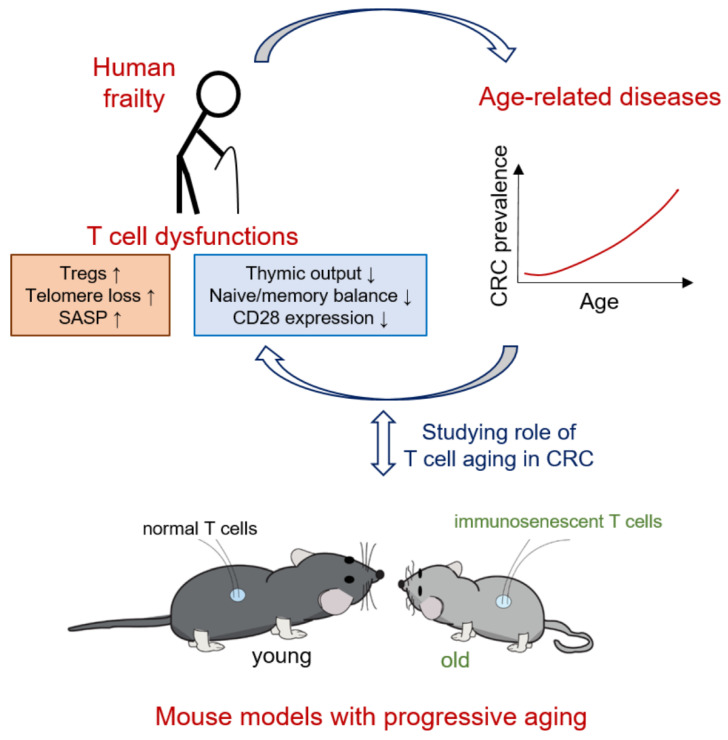
Aging affects the organism as a whole, inducing T cell dysfunctions as well. Increased (↑) numbers of Tregs, secretory-associated senescence phenotype (SASP) and telomere loss, as well as decreased (↓) thymic output, naïve/memory balance and CD28 co-stimulatory expression are specific signs of immunoaging. This can lead to the development of CRC, which is predominant in elderly. Using mouse models with progressive aging might elucidate the functional role of T cell aging in individuals with colorectal cancer.

**Table 1 cancers-13-06227-t001:** Impact of type, numbers, and activity of T cells on colorectal cancer development.

T Cell Subtype	Role in CRC	Reference
CD3+ T cells	High numbers correlated with overall better patient survival	[67,70]
Bystander CD8+ T cells	Show no chronic antigen exposure, no impact	[76,77]
Tumor reactive CD8+ T cells	Highly cytotoxic; frequencies vary amongst patients	[76]
Exhausted and TEMRA CD8+ T cells	Often tumor-reactive cells, have high potential as a target in immunotherapy due to high PD-1 expression	[78]
Th1 cells	Cytotoxic; high numbers correlated with overall better patient survival	[67,70,71]
Th2 cells	No specific impact	[72]
Th17	Increased numbers in later CRC stages are detrimental	[75]
Tregs	Increased numbers correlate with worse outcome, can become highly suppressive, Tregs show increased exhaustion marker expression	[72,79,80]
Tfh cells	High numbers correlate with better outcome	[71]
Th1-like Tfh cells	Newly described, highly cytotoxic	[80,81]
Th9, Th22	Tumor promoting effect	[75]

**Table 2 cancers-13-06227-t002:** Premature T cell aging in mouse models.

Mouse Type	Age When Phenotype Is Observed	T Cell Phenotypes	Reference
Wildtype	>20 months old	Decreased proliferation Memory phenotype Increased Treg numbers	[31,87]
TERC^−/−^	after multiple crossings	Loss of proliferative potential Development of lymphomas	[91,95]
TERT^−/−^	after multiple crossings	No data	[113]
Mice with hyper-long telomeres	any age	No data	[96,97]
Klotho^−/−^	9 weeks old	Thymus involution Correlated to T cell specific diseases in elderly (RA)	[99,103]
Ku70^−/−^		Normal T cell development Early lymphoma development	[105]
Ku80^−/−^		T cells arrested at early developmental stage Early lymphoma development	[105]
Vav1Cre^+/−^ × ERCC^−/fl^	6 months old	Memory T cell accumulation Senescence markers	[109]
IL10^−/−^	as early as week 3	Increased CD4+ CD8+ T cells in the gut Increased IFNγ Th1 cells Colitis development	[112]

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
