# Peer review of "T Cell Aging in Patients with Colorectal Cancer—What Do We Know So Far?"

_cancers, 2021, doi:10.3390/cancers13246227_

Round 1

Reviewer 1 Report

This Review Article recapitulates the current knowledge regarding aging and T-cells in colorectal cancer (CRC). Aging is a topic that has recently attracted a great deal of attention from the international community, not only in CRC but also in other malignancies; therefore, I found the Review very interesting and well-contextualized with the present knowledge.

The manuscript is written in a clear and well-organized manner; I appreciated the efforts of the Authors in conducting a systematic examination of the current literature on the field; another point of relevance is the last paragraph, where they examined the possible application of animal models to study the role of aged T cells in CRC.

In summary, this is an excellent work; in the light of improving its clarity and incisiveness, I would only suggest adding just another one or two figures that could enable the reader to better follow the discussion, as the T cells might not represent the primary field of research for a wide audience that is interested in aging in different cellular contexts. Finally, I would add some other details regarding the possibility of applying single-cell sequencing to study T-cells in CRC, if present in literature.

Author Response

We thank the reviewer for the kind and positive comments. We have provided another Figure which encompases the conclusions of the manuscript. Single-cell sequencing is indeed currently used to understand T cell behaviors in patients with CRC. We have now included the literature available. We have also now discuss the posibility of using sc-RNA seq as a method to evaluate T cell aging in patients with CRC.

Reviewer 2 Report

In this study, entitled “T cell aging in patients with colorectal cancer – what do we 2 know so far?”,  

Oana-Maria Thoma et al  reviewed the knowledge about age-related T cell dysfunctions in elderly patients with colorectal cancer and how aged T cells can mediate its development.

The review is well written, and the authors have extensively studied the literature.

 My minor comment is:

-line 37: please add a reference on that.

Author Response

We are grateful for the positive feedback on the manuscript. As per reviewer’s request, we have also added a reference on line 37.

Reviewer 3 Report

This review summarises the knowledge on T cell ageing and its potential role in colorectal cancer. The first 4 pages include detailed description of the process of T cell ageing that focus on T cell receptor diversity, naive and memory shift, loss of telomeres, apoptosis and cytokine production in the elderly. The Authors cite current literature necessary to understand changes in immune system of the old individuals. 

Next chapters describe role of T cells in humans with colorectal cancer and summarise some animal models used to elucidate influence of aged T cells on occurrence of colorectal cancer. The main concussions drawn in the literature are listed in two tables. 

The knowledge presented in the paper is valuable not only for basic science researchers but also for clinicians. Therefore, in my opinion, the review should be interesting for the readers of Cancers. One example is information on CD3  immune infiltrates in the tumor that may predict prognosis better than histology. On the other hand, the Authors showed that numbers of Th17 and Tregs in tumor microenvironment are not beneficial for oncological outcomes. In addition, they indicated that more predominant telomere loss was observed in individuals with colorectal cancer in comparison with healthy ones. These observations are important for further studies and possible clinical implications. 

In general, the paper is scientifically sound, written in good English and contain current and crucial data on T cell role in colorectal cancer. 

Only minor corrections might be recommended for the Authors:

  • line 48-49: The Authors stated that treatment of CRC was based on removal of cancerous polyps. More appropriately the mainstay of treatment of CRC is resection of malignant tumours or resection of the bowel with tumor/cancer. Removal of polyps should be treated as  prophylaxis of CRC or treatment in case of some cancerous polyps.
  • Some minor English corrections are needed: line 130 should be "correlated", line 263 should be "than histology", line 281 parenthesis (round bracket) is not needed 

Author Response

We gratefully thank the reviewer for the feedback and helpful suggestions for improvement. We have changed the text according to the reviewer’s comments. Furthermore, we have re-evaluated the manuscript to ensure that English grammar and overall text are according to the standards.

Reviewer 4 Report

The manuscript by Thoma, et al. discusses and summarizes the current knowledge of T cell aging in CRC patients.  It is well-written and describes in depth the functional roles of T cells and consequences of aging in the fight against cancer.  There are not much to comment, however, a very minor comment will be that Figure 1 is not discussed much in the text.  The figure may be self-explanatory, but it will be easier to follow if the text refers to the figure.  Overall, it is a very nice review. 

Author Response

We thank the reviewer for the positive comments on our manuscript. We agree that Figure 1 is only mentioned in the text, but not explained. Therefore, we have done some changes to the text in Section 2, to ensure that Figure 1 is thoroughly described. We hope that these modifications meet the reviewer’s standards.